# CogniPair - Dynamic LLM Matching Algorithm in Chaotic Environments: Mimicking Human Cognitive Processes for Relationship Pairing

## Abstract

Dating applications in the digital era have transformed how people connect, yet they often fall short in simulating the comprehensive character and fostering truly compatible relationships due to their reliance on quantitative data. This paper proposes a novel framework to simulate human characters by leveraging Large Language Models (LLMs) to enhance matchmaking by understanding the nuanced fabric of human personality and social connections. Traditional algorithms often lack the depth needed for personalized matchmaking, whereas LLMs offer sophisticated linguistic and cognitive capabilities to simulate a person and complicated personal decisions. Our framework introduces a multi-agent system comprising the Persona, Preference, and Dating Memory modules, allowing for dynamic and nuanced user interactions. This approach addresses the limitations of conventional LLM frameworks by capturing detailed personal attributes, updating preferences, and learning from past interactions. Our system enhances the relevance and effectiveness of match recommendations, focusing on emotional compatibility and shared values, providing a more personalized and responsive user experience in the dating domain.

## 1 Introduction

In the digital era, dating applications have become a prevalent medium for individuals seeking companionship and meaningful romantic relationships (1). These platforms utilize algorithms designed to match users based on shared characteristics and preferences, fundamentally transforming how people meet and interact. Despite their widespread adoption and technological advancements, a persistent critique remains: matches often lack the depth and understanding inherent in truly compatible human relationships (2). This issue underscores a critical gap in the matchmaking process—a reliance on quantitative data that fails to capture the nuanced complexities of human connections, such as individual personalities and the subtleties of social interactions.

Addressing this gap necessitates a paradigm shift in matchmaking technology, moving beyond traditional algorithms to embrace the intricacies of human preferences and interactions. Large Language Models (LLMs), such as GPT-3.5 and GPT-4 developed by OpenAI (3), represent a promising avenue for this shift. LLMs offer advanced capabilities in processing and generating natural language, mimicking human reasoning and understanding (4). Trained on extensive datasets covering a broad spectrum of human knowledge and interactions, these models can grasp subtleties and contexts in language previously unattainable by automated systems.

Integrating LLMs into dating applications introduces a novel approach to matchmaking. By leveraging their advanced linguistic and cognitive abilities, LLMs can analyze user data with a depth akin to human analysis. Unlike conventional matching algorithms that predominantly rely on quantitative analysis and fixed criteria, LLMs can interpret the complexities of user profiles and preferences, facilitating more meaningful and compatible connections. For instance, when users describe their interests or what they seek in a partner, LLMs can understand these descriptions beyond face value, capturing the broader context and nuances within their words (4). This deep understanding allows for matches based on personalities, preferences, and relationship goals, potentially leading to more fulfilling connections.

Moreover, LLMs can dynamically interact with user data, generating insights and predictions that evolve as more information becomes available (3). This adaptability enables LLMs to adjust to changes in users' preferences or circumstances over time, refining the matchmaking process to accurately reflect these developments.

However, despite their broad applicability, traditional LLMs exhibit significant limitations when applied to highly specific and personalized tasks like dating match recommendations (4). They often lack the granularity required for full personalization, failing to capture the nuanced preferences and dynamic nature of individual users' dating experiences. Typically designed to respond to general inquiries, these systems do not inherently focus on the unique requirements of the dating domain.

One major limitation is that traditional LLMs do not inherently account for individual differences in user preferences and personalities. Their responses tend to be generic, lacking the depth needed to tailor advice and match recommendations to each user's specific needs (5). Additionally, without a structured approach, LLMs interact with users in a static manner, unable to learn and adapt from ongoing interactions. This static nature restricts the models' ability to refine and personalize recommendations based on user feedback and evolving preferences. Furthermore, traditional LLMs are not specifically trained or optimized for the dating domain and lack the ability to understand and prioritize factors critical to successful romantic relationships, such as emotional compatibility, shared values, and personal growth dynamics (4).

To address these limitations, we propose a novel framework, **CogniPair**, which models each user as an autonomous sub-agent composed of interconnected modules: the **Preference Module**, the **Persona Module**, the **Reflection Module**, and the **Dating Memory Module**. This structured, multi-agent system allows for nuanced and dynamic interactions tailored specifically for the dating domain.

The **Persona Module** constructs a detailed profile for each user, capturing personal attributes such as personality traits, interests, and lifestyle preferences. The **Preference Module** gathers and updates user-specific preferences through initial questionnaires and feedback from interactions. The **Reflection Module** acts as the cognitive engine, integrating information from the Persona and Preference Modules, as well as past experiences, to simulate human-like decision-making. The **Dating Memory Module** stores records of past interactions, enabling the system to learn from experiences and adapt over time.

By integrating this framework, our contributions with this framework are:

- **Dynamic Cognitive Matching**: CogniPair introduces a novel matching algorithm that leverages LLM to simulate human cognitive processes in relationship pairing. This approach focuses on deeper aspects of matchmaking such as emotional compatibility and shared values, going beyond traditional reliance on quantitative data.

- **Modular Sub-Agent Architecture**: The system models each user as an autonomous sub-agent comprising several interconnected modules, including the Persona, Preference, Reflection, and Dating Memory Modules. This modular architecture allows for dynamic and personalized interactions, adapting to changes in user preferences and past interactions.

- **Human-Like Decision-Making**: The Reflection Module simulates human-like thinking, integrating user data and past interactions to refine decisions and improve match accuracy.

- **Realistic Interaction Simulation**: CogniPair generate realistic dialogues and personalities, ensuring emotionally aware and context-driven user interactions.

- **Iterative Learning and Adaptation**: Through the Dating Memory Module, the system stores past interactions, allowing continuous learning and refinement of preferences. This iterative process improves match suggestions over time, resulting in more relevant and satisfying pairings.

Our system aims to offer more relevant and effective match suggestions compared to general-purpose LLMs, thereby improving user satisfaction and the overall efficacy of digital matchmaking platforms.

## 2 LITERATURE REVIEW

Human dating activities involve numerous psychological aspects that current matching algorithms are unable to capture due to their complexity. Therefore, developing an algorithm capable of simulating these psychological facets is crucial for fostering meaningful romantic connections.

For instance, numerous studies, including the paper *"Big Five Personality Variables and Relationship Constructs,"* have demonstrated that personality traits such as openness, conscientiousness, extraversion, agreeableness, and neuroticism (the Big Five) play significant roles in romantic compatibility (6). Additionally, attachment theory suggests that individuals' attachment styles—secure, anxious, or avoidant—affect their relationship behaviors and expectations (7). Moreover, we incorporate a special agent in our architecture to simulate human emotion and conversation, engaging the concept of emotional intelligence, a widely studied topic. Emotional intelligence, which encompasses the ability to recognize and manage one's own emotions and the emotions of others, also contributes to relationship success (8).

By integrating the latest psychological studies on internal motivation into our persona-simulating architecture, we can simulate human-like personalities and their reactions using our reflection module. By incorporating these multifaceted psychological insights into dating algorithms, we can move beyond superficial matching criteria towards deeper, more meaningful compatibility.

Large Language Models (LLMs) like GPT-3.5 and GPT-4, developed by OpenAI, have made significant strides in simulating human personality (4). These models are trained on extensive datasets, allowing them to generate text that closely mimics human language patterns. The paper *"Editing Personality for Large Language Models"* has shown that LLMs can be fine-tuned to reflect specific personality traits and behavioral tendencies by analyzing linguistic cues (9). For example, by processing users' written profiles and interactions, LLMs can infer personality characteristics that are crucial for matching, such as introversion vs. extraversion or thinking vs. feeling (9; 4). This ability to simulate human personality with high fidelity makes LLMs powerful tools for creating personalized and psychologically informed matchmaking systems.

In addition to simulating personality, LLMs are adept at modeling social behavior, which is essential for predicting and enhancing interpersonal interactions in dating contexts. Studies have demonstrated that LLMs can be trained to recognize and predict patterns of social behavior by analyzing conversational data and user interactions (10). For instance, LLMs can identify common themes in how users communicate their interests, preferences, and emotions, enabling the creation of dynamic and context-aware matchmaking processes. This dynamic modeling allows the system to adapt to changes in users' social dynamics and preferences over time, providing more accurate and relevant match suggestions (11). By simulating social behaviors, LLMs can enhance the depth and quality of matches, facilitating connections that are not only based on shared interests but also on compatible interaction styles and emotional responses.

The integration of psychological principles and advanced LLM capabilities in matchmaking algorithms represents a significant advancement in the field of online dating. By leveraging the nuanced understanding of human personality and social behavior that LLMs offer, developers can create more effective and satisfying dating experiences for users.

## 3 METHODS

In this section, we present the framework of **CogniPair**. Our goal is to simulate human-like agents capable of reflective decision-making and iterative learning, thereby improving the quality and satisfaction of matches. The framework consists of several key components organized into a modular architecture, referred to as the **Sub-Agent Architecture**. We first provide an overview of the architecture, followed by detailed descriptions of each module and their interactions.

Figure 1 illustrates the overall architecture of our system, highlighting the flow of information between modules and the interactions among sub-agents.

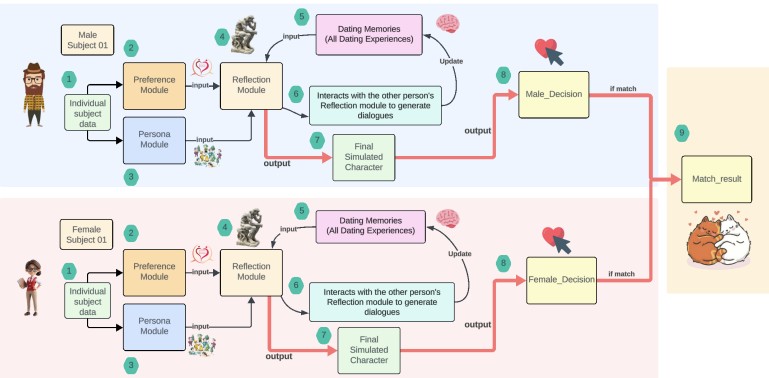

Figure 1: Overview of the Sub-Agent Architecture for Enhanced Matchmaking

## 3.1 DATA PREPROCESSING

For each individual user, we initiate the process by collecting their **individual subject data** (①). This comprehensive dataset encompasses personal information, personality traits, interests, values, lifestyle preferences, and desired attributes in a potential partner. This data is subsequently partitioned and refined into two primary components: **Preference Data** (②) and **Persona Data** (③). Preference Data includes the user's preferences regarding potential partners and relationships, such as desired attributes in a partner (e.g., age range, personality traits, interests) and relationship goals. Persona Data comprises all information related to the user's identity, including demographic details, personality traits, interests, values, and lifestyle preferences.

By segregating the data into these components, we create specialized modules capable of independently processing and updating user information, thereby enhancing the adaptability of the sub-agent.

## 3.2 SUB-AGENT ARCHITECTURE

The core of our **CogniPair** framework is the **Sub-Agent Architecture**, which models each user as an autonomous sub-agent composed of interconnected modules: the **Preference Module** (②), the **Persona Module** (③), the **Reflection Module** (④), and the **Dating Memory Module** (⑤). This architecture is designed to emulate human-like behavior and decision-making processes, allowing each sub-agent to independently learn and evolve based on individual experiences and interactions.

As depicted in our flowchart, the process begins by collecting the user's **individual subject data** (①), which includes personal information, personality traits, interests, values, lifestyle preferences, and desired attributes in a potential partner. This data is partitioned into the **Preference Module** (②) and the **Persona Module** (③). These modules provide prior information to the **Reflection Module** (④), acting as the cognitive engine of the sub-agent.

The Reflection Module interacts with the Reflection Modules of other sub-agents through simulated dialogues (⑥), facilitated by large language models (LLMs) such as GPT-4. These interactions are based on predefined themes (e.g., dating, first-time meeting), environments (e.g., restaurant), and time constraints (e.g., ten dialogue exchanges). The dialogues are stored in the **Dating Memory Module** (⑤), which functions as the long-term memory, recording details of the interactions.

After each interaction, the Dating Memory Module updates the Reflection Module, enabling the sub-agent to learn from experiences and adjust behavior and preferences accordingly. This iterative process continues as each sub-agent engages in speed dating with every other sub-agent, generating rich interaction data and refining their internal models.

Upon completion of all interactions, each sub-agent develops a **Final Simulated Character** (⑦) that encapsulates learned experiences and updated preferences. The **Matching Decision Mechanism** (⑧) is then employed, where each Reflection Module assesses the suitability of each dating partner, making a decision of **Yes** or **No**. A match is confirmed only if both sub-agents mutually

agree, ensuring reciprocal consent. The final matching results are generated based on these mutual decisions.

This comprehensive process, as illustrated in the flowchart, ensures that sub-agents effectively simulate the complexities of human dating, leveraging artificial intelligence to facilitate more meaningful and compatible matches. Details of some key modules will be introduced separately in the following sections.

The **Persona Module** constructs a comprehensive profile of the user, capturing attributes such as demographic information, personality traits, interests, values, and lifestyle preferences. This module leverages initial user inputs and may incorporate data from personality assessments to create a detailed persona representation.

The **Preference Module** captures the user's preferences regarding potential partners and relationships. It gathers information on desired attributes in a partner and relationship goals through user questionnaires and ongoing interactions.

The **Reflection Module** (④) acts as the cognitive engine of the sub-agent. It integrates prior information from both the Persona and Preference Modules, as well as past experiences stored in the Dating Memory Module. Utilizing large language models (LLMs) such as GPT-4, this module simulates human-like reflective thinking, enabling the sub-agent to make informed decisions and adapt over time. Key functions of the Reflection Module encompass **introspection**, which involves assessing the sub-agent's own persona and preferences to identify strengths and areas for growth; **interaction planning**, formulating strategies for interactions with potential matches considering the sub-agent's goals and past experiences; **decision-making**, evaluating potential matches and making decisions on pursuing relationships based on compatibility assessments; and **learning**, updating internal models and preferences based on interaction outcomes to facilitate continuous improvement.

The **Dating Memory Module** (⑤) functions as the long-term memory of the sub-agent, storing detailed records of past interactions and outcomes. This includes conversations, emotional responses, compatibility assessments, and feedback from previous dates. After each interaction, the Dating Memory Module is updated with new data, which is then used as input to update the Reflection Module. This continuous feedback loop enables the sub-agent to learn from experiences and adapt its behavior and preferences accordingly.

### 3.3 INTERACTION PROCESS

The interaction process involves sub-agents engaging in simulated dating scenarios. This process is facilitated by LLMs and is structured as follows: First, the **Dialogue Simulation** (⑥) occurs where the Reflection Module of one sub-agent interacts with the Reflection Module of another sub-agent. They engage in a conversation based on a predefined theme (e.g., dating, first-time meeting), environment (e.g., restaurant), and time constraint (e.g., ten dialogue exchanges). The LLM generates natural language dialogues, simulating a realistic interaction. Next, the dialogues are stored in each sub-agent's Dating Memory Module (⑤), recording the details of the interaction. The new data from the Dating Memory Module is then utilized to update each sub-agent's Reflection Module, allowing them to learn from the interaction and adjust their behavior and preferences. This process is iteratively repeated as each sub-agent participates in speed dating with every other sub-agent, generating rich interaction data and refining their internal models.

### 3.4 FINAL SIMULATED CHARACTER AND MATCHING DECISION

Upon completion of all interactions, each sub-agent has developed a **Final Simulated Character** (⑦) that encapsulates their learned experiences and updated preferences. The **Matching Decision Mechanism** is then employed, involving the Reflection Modules. Each sub-agent's Reflection Module assesses the suitability of each dating partner (⑧), making a decision of **Yes** or **No** for potential matching. A match is confirmed only if both sub-agents have a **Yes** decision for each other, ensuring reciprocal agreement. The final matching results are generated based on these mutual decisions. This approach ensures that matches are based on updated preferences and learned experiences, thereby increasing the likelihood of successful and satisfying matches.

## 4 EXPERIMENTAL SETUP

In this section, we introduce the specific datasets (Sec. 4.1) and models (Sec. 4.2) involved in our experiments. We also introduce the experimental designs of our experimental groups (Sec. 4.3), and the benchmarks (Sec. 4.4).

For this study, we utilized the Speed Dating Experiment dataset, which was compiled by Columbia Business School professors Ray Fisman and Sheena Iyengar for their research on gender differences in mate selection. The dataset includes information gathered from speed dating events conducted between 2002 and 2004. Participants in these events engaged in a series of four-minute "first dates" with individuals of the opposite sex, during which they rated their partners on six key attributes: Attractiveness, Sincerity, Intelligence, Fun, Ambition, and Shared Interests. At the conclusion of each date, participants indicated whether they would like to see their partner again.

In addition to these ratings, the dataset provides extensive questionnaire data collected at various stages of the process. This data encompasses demographic information, dating habits, self-perceptions across key attributes, beliefs about what others find valuable in a mate, and lifestyle details (12).

Essentially, we aimed to emulate participants' psychological profiles from their observed behaviors using GPT-4, capturing the nuanced aspects of their personalities and interactions. Through this setup, we sought to gain insights into the effectiveness of GPT-4 in replicating human behavior and to explore the attributes most influential in the initial stages of romantic interest, ultimately contributing to advancements in automated dating matching systems. The system outputs all identified matches in the format (male_id, female_id), representing the matched pairs.

### 4.1 SIMULATED SPEED-DATING PROCESS

Our primary objective was to assess the ability of using GPT-4 to simulate human character and interactions with a high degree of fidelity (13). By doing so, we aimed to evaluate the potential for creating human-like characters and interactions, which could inform the development of next-generation dating matching algorithms.

#### 4.1.1 PARTICIPANT SELECTION AND SIMULATION

To evaluate the performance of our model against human interactions, we incorporated both human-generated dialogues and GPT-4 simulated dialogues in our experiment. For the real dialogues, we invited 10 male and 10 female participants with the same range of age. These human participants engaged in real-time conversations, and their interactions were recorded and utilized as the **real dialogues** for our study. This approach allowed us to capture authentic human communication, providing a baseline for comparison.

For the simulation component, we replicated the personalities and interactions of 10 participants within a virtual environment using GPT-4. By carefully designing prompts and leveraging our framework, we generated virtual characters that mirrored the original participants' profiles. This setup enabled us to create **simulated dialogues** that closely resembled the human interactions in content and style.

By having both real and simulated dialogues from the same set of participants, we could directly compare the authenticity, realism, and consistency of GPT-4 generated interactions against human ones. This dual approach strengthened the validity of our evaluation metrics and provided deeper insights into the capabilities and limitations of our proposed framework.

#### 4.1.2 VIRTUAL ENVIRONMENT SETUP

The Speed Dating events were conducted in an enclosed room within a popular bar/restaurant near the campus. The table arrangement, lighting, and type and volume of music played were held constant across events. We tested the environment setup for virtual characters by replicating the original experiment through prompts to create a similar virtual environment.

We simulated the first round of the speed dating process involving the 10 male and 10 female participants using a large language model. To ensure consistency with the original experiment, we

simulated their speed dating procedure by allowing each virtual character to engage in a total of ten exchanges, replicating the 4-minute limit of their real-life interactions.

## 4.2 EXPERIMENTAL BENCHMARKS

To evaluate the efficacy of our proposed framework, we established three benchmark methods that represent typical approaches to the dating match task. These benchmarks serve as comparative standards against which we assess the performance of our primary method (Method 4). Below, we provide a detailed description of each benchmark method, including their design and implementation based on key performance metrics.

### 4.2.1 BENCHMARK 1: ONE-TO-ONE COMPATIBILITY SCORING

This foundational benchmark represents a standard matching algorithm used in dating applications (14), assessing compatibility through a weighted scoring system based on several attributes. The key attributes and their roles in determining compatibility include:

**Age Difference:** Preference for a minimal age gap, ideally five years or less. **Field of Study & Career:** Higher scores for similar fields of study and career paths. **Income Level & Social Activities:** Compatibility is enhanced when participants have similar income levels and social activity frequencies. **Shared Interests & Life Goals:** Significant score increases for shared interests (e.g., sports, arts) and aligned life goals (e.g., social motivations). **Key Personal Characteristics:** Attributes like attractiveness, sincerity, and intelligence are critically evaluated.

The algorithm calculates a compatibility score for each male-female pair by summing the weighted scores of these attributes. A match is recorded if this score surpasses a specific threshold, ensuring a substantial compatibility level between participants.

### 4.2.2 BENCHMARK 2: ZERO-SHOT MATCHING USING LLMS

In this method, we leverage Large Language Models (LLMs) to enhance the matching process through zero-shot learning. Each participant's detailed profile is input into an LLM (specifically GPT-4) without prior examples or fine-tuning related to matchmaking tasks. The LLM is prompted to identify the best possible matches based on shared values, interests, career fields, and lifestyle habits. The objective is to create meaningful connections that align with participants' personalities and goals. For each female participant, the model generates a list of potential male matches by analyzing the compatibility of profiles based on its pre-trained knowledge without additional supervision. This approach simulates an intuitive, manual matching process facilitated by the LLM's understanding of human language and relationships.

### 4.2.3 BENCHMARK 3: FEW-SHOT MATCHING WITH EXAMPLES USING LLMS

This method builds upon the previous approach by incorporating few-shot learning, where the LLM is provided with a few examples of successful matches to guide its matching process. By learning from these examples, the LLM can better identify meaningful connections between participants. In both Benchmark 2 and Benchmark 3, the use of LLMs allows for a more nuanced matching process that considers complex attributes and similarities between participants, potentially leading to more meaningful connections compared to traditional scoring methods. By comparing these benchmarks with our primary framework, we aim to demonstrate the advantages of incorporating a structured, multi-agent system with reflective capabilities in enhancing matchmaking outcomes.

## 5 EVALUATION METRICS

We evaluate the performance of our proposed approach using a set of carefully designed metrics intended to measure two main aspects: (1) the realism of the generated dialogues and the authenticity of the simulated characters (Sec. 5.1); and (2) the accuracy of matching decisions (Sec. A.1.1). Including standard deviations in our calculations provides insights into the variability of human assessments, which is crucial for understanding the consistency and reliability of subjective evaluations in human-computer interaction studies.

## 5.1 REALISM OF GENERATED CHARACTERS AND DIALOGUES

We designed a questionnaire to evaluate the realism and authenticity of the generated content, utilizing human evaluators for their unique perspectives on dialogue authenticity and character realism. These subtle aspects might be overlooked by automated systems (15). The key metrics used are:

**Dialogue Realism Score (DRS)**: Measures the average realism rating of the dialogues as perceived by human evaluators on a scale from 1 to 10. **Simulated Character Authenticity (SCA)**: Assesses the authenticity of the simulated characters based on human evaluator ratings, also on a scale from 1 to 10. **Simulated Compatibility Score (SCS)**: Evaluates the correlation between predicted compatibility scores and observed outcomes across participant pairs. **Persona Generation Accuracy (PGA)**: Calculates the accuracy of the generated personas by comparing generated attributes against ground truth values. **Persona-Dialogue Consistency Score (PDCS)**: Measures the semantic similarity between the generated personas and dialogues to assess consistency. The detailed formulas for these metrics are provided in Table 1 in the Appendix.

## 5.2 ASSESSMENT OF MATCHING DECISIONS

To evaluate the accuracy of the matching decisions made by the simulated agents, we use the following metrics: **Overall Accuracy (ACC)**: Measures the proportion of correctly identified matches and non-matches out of all possible cases. **True Positive Rate (TPR)**: Focuses on the system's ability to correctly identify genuine matches, which is crucial for the effectiveness of the matchmaking system. The detailed formulas for these metrics are also provided in Table 2 in the Appendix.

# 6 RESULTS

In this section, we present and analyze the results of our main method in comparison with other classical dating match experimental designs. We begin by evaluating the performance of our model in comparison with human participants, using the evaluation metrics defined earlier (Sec.4). Subsequently, we compare the matching results of our method with the benchmark methods, illustrating the improvements achieved by our proposed framework over traditional approaches (Sec.3).

## 6.1 EVALUATION OF MODEL AND HUMAN PERFORMANCE

To assess the effectiveness of our proposed framework, we compared the performance of GPT-4 generated dialogues and characters with those of human participants, using the evaluation metrics described in Section 5. The following table summarizes the scores obtained by both the model and human participants:

The **Dialogue Realism Score (DRS)** and **Simulated Character Authenticity (SCA)** show that human-generated dialogues and characters were perceived as more realistic and authentic compared to those generated by the model, with humans achieving higher scores in both metrics. This suggests that while GPT-4 can generate coherent dialogues, it may still lack the nuanced understanding of human emotions and social cues, which are essential for the perception of realism and authenticity.

Conversely, the model excelled in the **Simulated Compatibility Score (SCS)**, **Persona Generation Accuracy (PGA)**, and **Persona-Dialogue Consistency Score (PDCS)**, outperforming humans in these areas. This indicates the model's effectiveness in systematically analyzing data to predict compatible matches and maintain consistency between persona profiles and generated dialogues, demonstrating strengths in handling structured data and maintaining internal consistency critical for personalized matchmaking systems.

## 6.2 COMPARISON OF MATCHING RESULTS

In this section, we compare the results of our main method (Method 4) with other classical dating match experimental designs.

**Benchmark 1: One-to-Many Compatibility Scoring** — This foundational benchmark exhibited a moderate True Positive Rate (TPR) of 0.5484, indicating just over half of the matches were predicted

correctly. However, the Overall Accuracy (ACC) was relatively low at 0.2500, highlighting a considerable number of mismatches. This method serves as a baseline for understanding the effectiveness of more complex methods.

**Benchmark 2: Zero-Shot Matching** — Utilizing a zero-shot prompting approach, this method showed a lower TPR of 0.2581, reflecting its limited effectiveness in predicting accurate matches. The Overall Accuracy was also low at 0.1818, indicating a substantial number of incorrect and missed matches. While the zero-shot approach provides a straightforward method without specific training examples, it does not significantly enhance match accuracy.

**Benchmark 3: Few-Shot Matching** — Building upon Method 2, this method involved a few-shot prompting process which provided the model with a few examples of desired matches. It achieved a TPR of 0.2903 and an Overall Accuracy of 0.2000, indicating only marginal benefits from the few-shot prompting. This incremental improvement suggests that the few-shot approach aids in decision-making but is insufficient for substantial accuracy gains.

**Main Method: Simulated Character Matching Experiment** — Representing an advanced version of the previous methods, our primary method significantly enhanced the accuracy of predicting matches. It achieved the highest TPR of 0.6774 and an Overall Accuracy of 0.3182, underscoring its superior accuracy in predicting matches. This method demonstrates the most advanced and effective approach in our experiment.

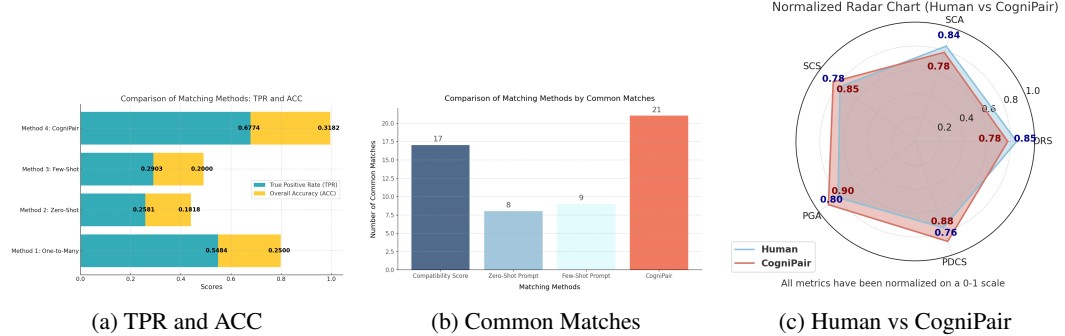

(a) TPR and ACC      (b) Common Matches      (c) Human vs CogniPair

Figure 2: Comparison of Matching Methods and Model Performance

# 7 FINDINGS

The findings from our experiment have several important implications for the field of match prediction and the broader application of artificial intelligence in social and relational contexts. The varying performance across the different methods highlights the potential and limitations of different AI-driven matching approaches.

Firstly, the relatively low performance of the Zero-Shot Matching method (Benchmark 2) underscores the challenges of accurately predicting matches without domain-specific training data and proper matching method. This suggests that while zero-shot approaches are useful for initial baselines, they are not sufficient for applications where nuanced understanding and contextual reasoning are critical. The improvement seen in the Few-Shot Matching method (Benchmark 3) indicates that providing the model with examples can enhance its predictive capabilities, albeit marginally. This points to the importance of example-based learning in fine-tuning AI systems for specific tasks, suggesting that few-shot learning may serve as a useful bridge between zero-shot and fully supervised methods.

The significant improvements observed in the Simulated Character Matching Experiment (Main Method) highlight the benefits of incorporating more sophisticated AI techniques, such as simulated interactions and character modeling. This method's higher True Positive Rate and Overall Accuracy indicate that AI systems can better predict human-compatible matches by simulating deeper, more nuanced interactions. This suggests a potential pathway for developing more advanced AI systems that not only match based on static attributes but also consider dynamic and interactive elements that reflect real-world relational dynamics.

Moreover, the success of the Simulated Character Matching Experiment implies that AI systems can be designed to mimic human-like decision-making processes more effectively, providing insights into how people might perceive compatibility. This has broader implications for AI applications beyond matchmaking, such as in customer service, therapeutic contexts, and educational tools, where understanding and mimicking human interactions can enhance user experiences and outcomes.

The findings also emphasize the need for further research into optimizing AI models for complex, real-world tasks. The relatively modest overall accuracy across all methods indicates that there is still substantial room for improvement. Future work could explore the integration of more advanced techniques such as reinforcement learning, contextual embeddings, or multi-modal data inputs, which could provide richer and more accurate predictive capabilities.

In summary, the experiment underscores the importance of advanced AI techniques in enhancing the accuracy and reliability of match prediction systems. It highlights both the challenges and opportunities in the field, suggesting a promising direction for future research and development. The use of simulated character interactions, in particular, emerges as a powerful tool for improving AI's ability to understand and predict human preferences and behaviors.

# 8    LIMITATIONS OF THE EXPERIMENT

One of the primary limitations of our experiment is the sample size. We only assessed 10 male and 10 female participants, which may not fully represent the diversity and complexity of real-world dating scenarios. This small sample size can lead to skewed results and limit the generalizability of our findings. Additionally, the dataset's homogeneity might not capture the wide range of personalities and behaviors found in a larger, more diverse population. Another limitation is the reliance on simulated dialogues and profiles, which, despite our best efforts, might not perfectly replicate real human interactions. While our evaluation metrics suggest that the generated dialogues are quite realistic, subtle nuances of human behavior might still be missing.

Several factors could contribute to the disadvantages observed in our experiment. Firstly, the complexity of human emotions and interactions poses a significant challenge for any AI-driven matching system (2). Our models, although sophisticated, might not fully capture the intricacies of human relationships. Secondly, the token limits of LLMs constrain the amount of data processed simultaneously, potentially affecting the depth of analysis(16). To mitigate this, we employed LLM dynamic coding, which allows for more efficient processing, but this approach is still in its early stages and may require further refinement.

Furthermore, the appearance and physical attributes of individuals, such as facial features and body type, play a crucial role in initial attraction and decision-making in real-world dating scenarios(17). Hormonal influences and visual appeal significantly impact one's choice to pursue a potential partner. Our dataset, however, lacks any facial or body data, which limits the ability to incorporate these important aspects into our analysis.(17) This omission likely affects the realism and accuracy of the simulated matches, as physical appearance is a critical factor in human mate selection.

Lastly, the integration of multiple agents and modules, while beneficial for a multi-dimensional analysis, introduces additional complexity and potential points of failure. Ensuring seamless collaboration among these agents is crucial for the system's overall effectiveness.

# 9    FUTURE DIRECTIONS

Looking forward, we aim to address the limitations identified in this study. Expanding the sample size and incorporating a more diverse dataset will be crucial for validating our findings and improving the robustness of our models. Additionally, refining our LLM dynamic coding approach and enhancing the integration of multi-agent systems will further optimize the matchmaking process.

Our ultimate goal is to bridge the gap between algorithmic precision and the complex nature of human relationships, creating a dating application that fosters genuine connections. By continuing to innovate and improve our methods, we hope to contribute to a new era of technology-driven matchmaking that is deeply rooted in a nuanced understanding of human emotions and behaviors.

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

## A  APPENDIX

### A.1  EVALUATION METRICS

These measures provide a detailed evaluation of the generated content's authenticity, realism, and alignment with human perceptions and actual outcomes.

#### A.1.1  ASSESSMENT OF MATCHING DECISIONS

To evaluate the accuracy of the matching decisions made by the simulated agents, we employ the metrics summarized in Table 2.

Here, $T_p$ is the number of true positives (correctly identified matches), $T_n$ is the number of true negatives (correctly identified non-matches), $F_p$ is the number of false positives (incorrectly identified matches), and $F_n$ is the number of false negatives (missed matches).

In the context of our task—finding suitable dating matches—the True Positive Rate (TPR) is particularly significant because our primary goal is to identify and match individuals who are genuinely compatible. Focusing on TPR helps ensure that the system accurately recognizes and prioritizes potential matches, leading to a more effective and satisfying matchmaking experience.

Table 1: Summary of Realism and Authenticity Metrics

| Metric | Formula and Description |
|---|---|
| Dialogue Realism Score (DRS) | $\text{DRS} = \dfrac{1}{N} \sum_{i=1}^{N} \left( \dfrac{1}{K} \sum_{k=1}^{K} r_{ik} \right)$ 
 Averaging realism ratings $r_{ik}$ from $K$ evaluators over $N$ dialogues. 
 $\sigma_{\text{DRS}} = $ Standard deviation across $N$ dialogues and $K$ evaluators |
| Simulated Character Authenticity (SCA) | $\text{SCA} = \dfrac{1}{M} \sum_{j=1}^{M} \left( \dfrac{1}{L} \sum_{l=1}^{L} a_{jl} \right)$ 
 Averaging authenticity ratings $a_{jl}$ from $L$ evaluators over $M$ characters. 
 $\sigma_{\text{SCA}} = $ Standard deviation across $M$ characters and $L$ evaluators |
| Simulated Compatibility Score (SCS) | $\text{SCS} = \dfrac{\sum_{p=1}^{P}(s_p - \bar{s})(o_p - \bar{o})}{\sqrt{\sum_{p=1}^{P}(s_p - \bar{s})^2 \sum_{p=1}^{P}(o_p - \bar{o})^2}}$ 
 Pearson correlation between predicted compatibility scores $s_p$ and observed outcomes $o_p$ across $P$ participant pairs. |
| Persona Generation Accuracy (PGA) | $\text{PGA} = \dfrac{1}{S} \sum_{s=1}^{S} \left( 100\% - \text{MSE}_s \right)$ 
 Where $\text{MSE}_s = \dfrac{1}{n} \sum_{i=1}^{n} (y_{si} - \hat{y}_{si})^2$ for each persona $s$ and its attributes. |
| Persona-Dialogue Consistency Score (PDCS) | $\text{PDCS} = \dfrac{1}{U} \sum_{u=1}^{U} s_u$ 
 Evaluates semantic similarity scores $s_u$ between generated personas and dialogues over $U$ dialogues. |

Table 2: Summary of Matching Decision Metrics

| Metric | Formula and Description |
|---|---|
| Overall Accuracy (ACC) | $\text{ACC} = \dfrac{T_p + T_n}{T_p + T_n + F_p + F_n}$ 
 Measures the proportion of correctly identified matches and non-matches out of all cases. |
| True Positive Rate (TPR) | $\text{TPR} = \dfrac{T_p}{T_p + F_n}$ 
 Focuses on the system's ability to correctly identify genuine matches, crucial for matchmaking effectiveness. |

