# OpenReview forum: "CogniPair - Dynamic LLM Matching Algorithm in Chaotic Environments Mimicking Human Cognitive Processes for Relationship Pairing"
_ICLR.cc/2025/Conference — ICLR 2025 Conference Withdrawn Submission_

### Official Review · Reviewer_ByNt · 2024-10-27

**Soundness:** 3
**Presentation:** 3
**Contribution:** 3
**Rating:** 8
**Confidence:** 3

**Summary:**

The paper introduces CogniPair, a novel framework that leverages Large Language Models (LLMs) to simulate human cognitive processes for improved matchmaking in dating applications. Recognizing the limitations of traditional algorithms that rely on quantitative data and fail to capture the complexities of human personality and social interactions, CogniPair employs a multi-agent system comprising the Persona, Preference, Reflection, and Dating Memory modules. This architecture allows for dynamic, personalized interactions by capturing detailed personal attributes, updating preferences, and learning from past experiences. Experiments using the Speed Dating Experiment dataset demonstrate that CogniPair outperforms traditional matching algorithms, enhancing match accuracy by focusing on emotional compatibility and shared values. Despite limitations like a small sample size and challenges in fully replicating human nuances, the framework shows promise for advancing AI-driven matchmaking by providing more meaningful and satisfying connections.

**Strengths:**

Introduces a novel approach to matchmaking by leveraging Large Language Models (LLMs) to simulate human cognitive processes, moving beyond traditional algorithms that rely solely on quantitative data. Potentially impactful and many future work.

Incorporates insights from psychological studies on personality traits and internal motivations to enhance the realism of simulated human-like personalities and interactions.

Shows promise for advancing the field by providing a more sophisticated, AI-driven approach to matchmaking that better reflects the complexities of human social interactions.

**Weaknesses:**

1.Human evaluation could be conducted to see if the strategies indeed helps.

2.Multiple environments should be tested based on personality

**Questions:**

How do you envision this work impacts future research in relationship pairing and dating?

---

### Official Review · Reviewer_1cVz · 2024-10-31

**Soundness:** 3
**Presentation:** 1
**Contribution:** 2
**Rating:** 3
**Confidence:** 2

**Summary:**

This paper proposes the CogniPair framework, which uses a dynamic LLM matching algorithm to simulate human cognitive processes to improve the quality of dating pairing. Through a modular sub-agent architecture, the model can achieve a deeper level of emotional compatibility and matching of shared values, surpassing the quantitative analysis of traditional algorithms. The idea is novel and the topic is interesting, but the overall work is not solid enough.

**Strengths:**

The research perspective is quite innovative. Not only the original matching conditions are taken into account, but also the degree of compatibility after simulated interaction.

**Weaknesses:**

The work is not solid enough. The number of data sets, the number of comparison methods and the number of test samples are not enough. The discussion experiments also need to be increased, otherwise it is difficult to determine whether the effect of the experiment is accidental.

**Questions:**

1.	As the author said, the sample size is too small, with only ten men and ten women, and the metrics in the evaluation process are highly dependent on human subjectivity. It is recommended to increase the sample size or add other objective metrics.
2.	There is only one dataset. The preference module needs to rely on questionnaires. Does this limit the data size?
3.	The detailed parameters of the dataset are not listed, so it is impossible to see the impact of data sparsity on the model.
4.	Too few bench lines. There are only three bench lines. It is recommended to add more results from the past three years to reflect the superiority of the method proposed in the article.
5.	Regarding the accuracy of matching decisions, please explain why ACC and TPR are chosen.
6.	There are no method formulas in the whole article, which makes all the method details in the article unclear. I hope to show the specific method flow through some formulas.
7.	The experiment is weak. There is no ablation experiment to prove the effectiveness of the reflection module and the dating memory module, and there is no experiment to discuss the optimal number of dialogue iterations, etc. More experiments are needed to illustrate the necessity and performance of the module.

---

### Official Review · Reviewer_AAvZ · 2024-11-02

**Soundness:** 1
**Presentation:** 1
**Contribution:** 1
**Rating:** 3
**Confidence:** 4

**Summary:**

This manuscript presents a novel framework for enhancing matchmaking by using LLMs to simulate human characters. The framework is based on a multi-agent system that includes various modules (namely Persona, Preference, and Dating Memory) aimed at capturing various nuances of human personalities and social connections. The goal of this paper is to mimic human cognitive processes and improve match recommendations.

**Strengths:**

LLMs have been proven effective in mimicking human behaviors under certain conditions, and the versatility of a multi-agent system can contribute to enhancing matchmaking strategies and recommendation.

**Weaknesses:**

This manuscript presents several criticalities that undermine its overall quality. These include:
- First, this paper is strongly descriptive and not technically dense. There is no formal presentation of the framework's modules. Furthermore, how these are interconnected and interoperable is not discussed. This trend extends to all key sections: the data preprocessing step is unclear and ambiguous, the sub-agent architecture is superficially discussed and not detailed, and the interaction/decision process is not adequately described.
- The experimental section is verbose and definitely unclear: the authors first present an existing dataset (i.e., Speed Dating Experiment) for simulating behaviors, then introduce real-life interactions based on (only) 20 participants, plus simulated dialogues to compare the reliability. The experimental setup is also heavily fragmented and not adequately discussed. The assessment criteria are not properly contextualized (just summarized in a table in the Appendix).
- Shown results turn out to be poor and no comparison with baseline or competing methods is shown.
- The presentation of this manuscript has much room for improvement. Papers are cited by their title (see rows 114 and 130), figures are not properly readable, phrases are repeated multiple times, and the overall discussion is verbose, requiring proofreading and massive rephrasing.
- The participants' recruitment and data treatment is not discussed at all and must be properly detailed.

**Questions:**

- How are the capabilities of individual modules validated? For instance, how does the Persona module shape traits and how are these assessed?
- How is the Dating Memory Module implemented?
- How are agents deployed?
- What are the parameters of the used LLM (e.g., temperature, top_p, top_k)?
- Based on which criteria were participants recruited? How were their interactions recorded and handled?
- Why do the authors rely only on GPT-4 rather than experiment also with other models?
- What are male_id, female_id? Are these from the Speed Dating dataset?

**Details Of Ethics Concerns:**

The authors mention human participants for the experimental scenario (see row 303), yet no adequate discussion of their recruitment (e.g., informed consent) and data handling (e.g., data collection/storing/processing) is present.

---

### Official Review · Reviewer_KiKL · 2024-11-03

**Soundness:** 1
**Presentation:** 3
**Contribution:** 1
**Rating:** 3
**Confidence:** 3

**Summary:**

The paper proposes CogniPair, a matchmaking framework that leverages large language models (LLMs) for simulating human cognitive processes in pairing relationships within dating applications. It presents a modular architecture with sub-agents, including the Persona, Preference, Reflection, and Dating Memory modules, designed to enhance matchmaking through simulations of human-like decision-making and learning based on past interactions. Experiments include simulations with a small dataset using GPT-4 and comparisons against baseline methods.

**Strengths:**

1. The modular approach with sub-agents seems well-conceived, enabling distinct functionalities to simulate human-like interactions.
2. The benchmarks including zero-shot and few-shot learning benchmarks provides a useful context for evaluating the framework’s efficacy over conventional approaches.

**Weaknesses:**

1. Critical implementation specifics, such as the prompting process for each sub-agent module, remain ambiguous. Further, there is insufficient information on Dating Memory Modules and how it updates the Reflection Module.
2. The experiments are confined to a small dataset from a speed-dating event with only 10 participants per gender, of the same age range, within the same environment. The generalizability of the results is questionable, as the dataset is insufficient to capture the complexities of diverse real-world dating preferences and settings.
3. The system's evaluation metrics (e.g., dialogue realism, character authenticity) rely heavily on subjective human ratings rather than objective, real-world testing on a dating app platform.
4. The paper claims that the Cognitive Matching and Reflection Modules capture emotional compatibility, yet the empirical evidence (DRS and SCA scores) suggest the opposite.

**Questions:**

Addressing the weaknesses,
1. Can you provide more concrete details on the sub-agent modules and their interactions? This would be help with reproducibility and provide a better understanding of the CogniPair system.
2. How would you address the limitations of the small dataset, and possibly extend it more diverse dating settings?
3. The paper would benefit if some examples outputs from the CogniPair's matchmaking are included.
4. Can you explain how the Cognitive Matching and Reflection Modules capture emotional compatibility, possibly through example outputs? Furthermore, how do you explain the discrepancy with the empirical evidence showing a lack of human emotion understanding?

---

### Note · Authors · 2024-12-12

I have read and agree with the venue's withdrawal policy on behalf of myself and my co-authors.